# Microstructure and Composition Evolution of a Fused Slurry Silicide Coating on MoNbTaTiW Refractory High-Entropy Alloy in High-Temperature Oxidation Environment

**DOI:** 10.3390/ma13163592

**Published:** 2020-08-14

**Authors:** Jiesheng Han, Bo Su, Junhu Meng, Aijun Zhang, Youzhi Wu

**Affiliations:** 1School of Materials Science and Engineering, Lanzhou University of Technology, Lanzhou 730050, China; jshan@licp.cas.cn; 2State Key Laboratory of Solid Lubrication, Lanzhou Institute of Chemical Physics, Chinese Academy of Sciences, Lanzhou 730000, China; bosu@licp.cas.cn (B.S.); zhangaijun@licp.cas.cn (A.Z.)

**Keywords:** silicide coating, refractory high-entropy alloys, Si-20Cr-20Fe, high-temperature oxidation, fused slurry method

## Abstract

In this paper, the Si-20Cr-20Fe coating was prepared on MoNbTaTiW RHEA by a fused slurry method. The microstructural evolution and compositions of the silicide coating under high-temperature oxidation environment were studied. The results show that the silicide coating could effectively prevent the oxidation of the MoNbTaTiW RHEA. The initial silicide coating had a double-layer structure: a high silicon content layer mainly composed of MSi_2_ as the outer layer and a low silicon content layer mainly contained M_5_Si_3_ as the inner layer. Under high-temperature oxidation conditions, the silicon element diffused from the silicide coating to the RHEA substrate while the oxidation of the coating occurred. After oxidation, the coating was composed of an outer oxide layer and an inner silicide layer. The silicide layer moved toward the inside of the substrate, led to the increase of its thickness. Compared with the initial silicified layer, its structure did not change significantly. The structure and compositions of the oxide layer on the outer surface strongly depended on the oxidation temperature. This paper provides a strategy for protecting RHEAs from oxidation at high-temperature environments.

## 1. Introduction

Refractory high-entropy alloys (RHEAs) are considered to be a new generation of high-temperature materials, because they have the advantages of both high-entropy alloys (HEAs) and refractory metals (RMs), such as high-temperature strength, high hardness, and good phase stability at high temperatures [1,2,3,4,5,6,7]. They are mainly composed of Mo, Nb, Ta, and W, while Ti, Cr, V, Si, and Al are usually used as strengthening alloy elements [8,9,10,11,12,13,14]. For example, Mo_25_Nb_25_Ta_25_W_25_ and Mo_20_Nb_20_Ta_20_W_20_V_20_ are two of the types of most extensively studied RHEAs [7]. The neutron diffraction analysis of those alloys after annealing at 1400 °C for 19 h shows that no changes have occurred to their phase structures. Their compression yield strengths are much higher than that of Inconel 718 alloy at temperatures above 800 °C. Furthermore, their yield strengths are still higher than 500 MPa at 1200 °C [15,16]. However, the poor oxidation resistance of RHEAs is a major obstacle for their use in high-temperature engineering applications, as are RMs. Successful applications of RMs in various turbines have suggested that their service lives may largely depend on their high-temperature oxidation resistance rather than their high-temperature mechanical properties [17]. Therefore, the improvement of their oxidation resistance is a prerequisite for the successful application of RHEAs in high-temperature oxidation environments.

The addition of anti-oxidant alloy elements is one of the strategies to improve the oxidation resistance of RHEAs, but it inevitably diminishes the mechanical properties. It is necessary to find a balance between the high-temperature mechanical properties and the oxidation resistance [18]. In the past years, more than 20 types of RHEAs have been studied to enhance their oxidation resistance through anti-oxidant alloy addition. The alloying elements enable the formation of protective oxide layers at elevated temperatures [19,20,21,22,23]. Gorr et al. reported two equiatomic AlCrMoTi-X (X = Nb or Ta) RHEAs [18,24]. In their work, the RHEAs sheets were exposed to air at temperatures from 900 °C to 1000 °C for 48 h. The results showed that the mass of the RHEAs increased less than those of RMs and TaAlCrMoTi RHEA had the superior anti-oxidant property. Other studies have also showed that the addition of Ti and Si is beneficial to the oxidation resistance of RHEAs [20,21,22,24].

Preparing anti-oxidation coatings on surfaces is another effective method to improve the oxidation resistance of RMs, allowing them to be successfully applied in engineering applications under high-temperature oxidation environments. Among anti-oxidation coatings, silicide coatings have been widely used to protect RMs and many studies have addressed on the Si-20Cr-20Fe coating. The technical report (AFML-TR-68-210) of the Air Force Materials Laboratory (USA) opened up the possibility of protecting RMs with fused silicide coatings [25]. It suggested that the Si-20Cr-20Fe coating was very stable in air at 1360 °C, and it was considered as the best candidate for protecting niobium alloys. The Si-20Ti-10Mo and Si-20Cr-1/2B_4_Si coatings were identified as better choices for tantalum and molybdenum alloys, respectively, because they can be reliably used at 1760 °C. The technical report of the NASA Contract (NGR-27-003-001) characterized the microstructure of the Si-20Cr-20Fe coating on a niobium alloy [26]. The technical report (NASA TN D-7617) evaluated the protective performance of the Si-20Cr-20Fe coating on three different RMs (FS-85, C-129Y, and Cb-752) in stagnation model plasma arc tests [27]. The technical report (NASA TN contractor report 201753) reported that the Si-20Cr-20Fe coating provided very good oxidation protection for Mo-Re alloys at 1260 °C and Mach 4 in the Hypersonic Materials Environmental Test System at the NASA Langley Research Center [28]. The Si-20Cr-20Fe coating had very stable performance in a combustion atmosphere at 1360 °C and could withstand hundreds of thermal shocks. It has been successfully used as a high-temperature protective coating for niobium alloy nozzles of F100 rocket engines [29]. Sankar et al. analyzed the microstructure and composition evolution of the Si-20Cr-20Fe coating on C103 alloy during oxidation and explained the formation and oxidation resistance mechanisms of the silicide coating [30].

The aim of this paper is to explore the feasibility of the Si-20Cr-20Fe coating in improving the oxidation resistance of MoNbTaTiW RHEA. The Si-20Cr-20Fe coating was prepared on the surface of MoNbTaTiW RHEA by a fused slurry method. To show the anti-oxidation mechanism of the silicide coating, the evolution of the structure and compositions of the coating at 1000 °C and 1300 °C were studied.

## 2. Materials and Methods

MoNbTaTiW RHEA with an equimolar ratio of the elements was prepared by spark plasma sintering. Commercially available Mo, Nb, Ta, Ti, and W elemental powders were used as raw materials. The purity of all the powders was higher than 99.5%. The particle size of Mo, Nb, Ta, and Ti were less than 38 µm and that of W was less than 5 µm. After being proportioned in an equal molar ratio, the powders were mixed by planetary ball milling for 6 h. The mixed powders were put into a graphite mold and sintered in a spark plasma sintering furnace by vacuum hot pressing. The sintering temperature was 1500 °C, the pressure was 30 MPa, and the holding time was 20 min. The as-prepared MoNbTaTiW RHEA was used as the substrates. The MoNbTaTiW RHEA was cut into 5 mm × 5 mm × 20 mm cuboids whose surfaces were polished using 2000 mesh sandpaper and cleaned with acetone.

A Si-20Cr-20Fe (wt.%) coating was prepared on the RHEA cuboids by a fused slurry method. The particle size of Si powder was less than 10 µm, and that of Cr powder and Fe powder was less than 76 µm. The purity of those powders was higher than 99.5%. First, the Si-20Cr-20Fe mixed powder was uniformly milled. Then, a polyethylene alcohol aqueous solution with a concentration of 4% that was used as a binder was added to the slurry. Subsequently, the RHEA cuboids were immersed in the slurry and lifted at a fixed speed, followed by drying and solidification at 80 °C in an oven. Finally, the RHEA cuboids coated with slurry were placed in a vacuum oven under a 5 × 10^−3^ Pa pressure and held at 1430 °C for 1 h to obtain a silicide coating. Table 1 presents the compositions and properties of the as-prepared MoNbTaTiW RHEA. The flowchart of the preparation process of samples and the XRD diffraction patterns of the RHEA and the coating slurry are shown in Figure 1. It can be seen that the RHEA has a single BCC phase and the coating slurry is a mixture of Si, Cr, and Fe.

For oxidation tests, the coated RHEA cuboids were cut into cube slices with a size of 5 mm × 5 mm × 5 mm. The side surfaces of the cube slices were bare MoNbTaTiW RHEA without the silicide coating. The cube slices were placed in an alumina crucible and positioned in a box furnace. The experimental parameters of oxidation were as follows; the heating rate was 10 °C/min, the oxidation temperature was set to 1000 °C and 1300 °C, the oxidation time was 1 h, and static atmosphere.

The crystal structures were determined by X-ray diffraction (XRD, Empyrean, Almelo, The Netherlands). The surfaces and structures of the silicide coating were observed using scanning electron microscopy (SEM, JSM-5600LV, Tokyo, Japan). The chemical compositions of the samples were examined by energy dispersive spectroscopy (EDS, X-MaxN, Abingdon, Oxon, UK).

## 3. Results and Discussion

### 3.1. Morphology and Microstructure of the Silicide Coating

Figure 2a shows the surface morphology of the fused slurry silicide coating prepared on MoNbTaTiW RHEA. The coating had a rough surface on which holes and cracks can be found. Figure 2b shows its XRD pattern. As can be seen, the coating is composed of disilicide (MSi_2_, M = W, Ti, Mo, and Ta), ternary silicide (Cr_4_Nb_2_Si_5_), and lower silicide (M_5_Si_3_, M = Ti and Ta).

Figure 3 shows the cross section of the silicide coating and the EDS elemental mappings of Si, Cr, and Fe. It can be seen that the silicide coating was ~113 µm thick. According to the distribution of silicon content, the silicide coating had a double-layer structure: a thicker outer layer that contained a higher content of silicon and a thinner inner layer that had a lower content of silicon. For example, the average atomic percentages of Si in regions A and B in Figure 3 were 66.2% and 37.9%, respectively. However, it should be noted that the contents of both Cr and Fe in the inner layer are much higher than those in the outer layer. It also can be seen that some cracks propagated from the surface to the inner layer of the silicide coating and even to the interface between the silicide coating and the RHEA substrate.

Previous investigations of silicide coatings on niobium alloys and tantalum alloys have shown that the outermost layers of the silicide coatings are disilicide layers, and the inner layers are low silicide layers [30,31,32]. In disilicide MSi_2_ and low silicide M_5_Si_3_, the theoretical percentages of Si atoms are 66.7% and 37.5%, respectively. Therefore, it can be inferred that the outer layer of the silicide coating consisted of disilicide (MSi_2_) while the inner layer was composed of lower silicide (mainly M_5_Si_3_), because silicidation and diffusion reactions occurred on the surface of the MoNbTaTiW RHEA substrate. This result is consistent with the XRD result as shown in Figure 2.

### 3.2. Oxidation Behavior of the Silicide Coating

Figure 4 shows the morphologies of the silicide coating after oxidation at different temperatures. After oxidation at 1000 °C, the surface of the silicide coating had characteristics of solidification after melting and oxide crystallization. The silicide coating became brown and was quite rough, and many cracks were generated on its surface. The uncoated side surfaces were oxidized to form a yellow oxide scale (Figure 4a). After oxidation at 1300 °C, the silicide coating became gray and an extremely thick brown oxide coating was formed on the uncoated side surfaces. The surface of the silicide coating was covered with granular crystals Figure 4b.

Figure 5 shows the XRD patterns of the surfaces of the silicide coating after oxidation at different temperatures. As can be seen, after oxidation at 1000 °C, the surface of the coating consisted of SiO_2_, TiO_2_, Cr_2_SiO_4_, Ta_0.3_W_0.7_O_2.85_, and Ti_0.67_Nb_1.33_O_4_ (Figure 5a). After oxidation at 1300 °C, the surface coating was composed of CrNbO_4_, SiO_2_, WO_3_, and Fe_2_SiO_4_ (Figure 5b).

Figure 6 shows the cross-section of the silicide coatings after oxidation at 1000 °C and 1300 °C, and the EDS mapping of the oxidized silicide coatings is shown in Figure 7. As shown in Figure 6, all the oxidized silicide coatings had a multilayered structure. The increases in the thickness of the coating were due to the continued diffusion reaction of Si, Cr, and Fe elements to the RHEA substrate at high temperatures. All the elements of the outer layer suffered from oxidation. Both the oxidized silicide coatings maintained good integrity, but the average thickness of the coatings increased significantly during the oxidation process. The thicknesses of the silicide coating oxidized at 1000 °C was 347 µm and that was about 397 µm after oxidation at 1300 °C, respectively. Based on the XRD analysis result of the surfaces after oxidation (Figure 5) and the EDS mapping, the outer layer was an oxide layer and the inner layer was a silicide layer. The coating oxidized at 1300 °C was thicker than the one oxidized at 1000 °C because of the higher diffusion rate of silicon, and the silicon had a deeper diffusion distance at the higher oxidation temperature [33].

However, there were significant differences in the structures of the coatings after oxidation at 1000 °C and 1300 °C. Figure 6a shows that the oxide layer exhibited a bulge phenomenon, and a gap appeared between the oxide and silicide layer after oxidation at 1000 °C. Many holes and cracks were found in the silicide layer. The cracks were originated from the interface between the silicide and oxide layers and terminated at the interface between the silicide layer and REHA substrate. According to the XRD analysis of the coating surface, the oxide scale was composed of silicon dioxide and complex metal oxides. The swelling of the oxide layer may have been caused by volume expansion during oxidation. The pores formed in the silicide layer were resulted from the diffusion of a large number of silicon atoms into the RHEA substrate [34].

After oxidation at 1300 °C, although the thickness of the coating increased, it still had a double-layer structure containing oxide and silicide layers, as shown in Figure 6b. Compared with the coating oxidized at 1000 °C, there was no gap existed between the oxide and silicide layers, though there are also many holes and cracks were formed in the silicide layer. According to the XRD analysis of the outer surface of the coating, the oxide layer mainly consisted of complete oxides of silicon and other metal elements, but no oxides of Mo and Ta were detected. The complete oxidation of the metal and silicon could cause a significant increase in the oxide layer volume, whereas the volatilization of some oxides (such as MoO_3_) could cause the loss of the oxide layer and reduce the volume [34]. Consequently, the volume of the oxide layer was basically the same as that of the initial silicide layer. Therefore, no bulge phenomenon or cracks appeared between the oxide layer and the silicide layer.

Comparing Figure 4 and Figure 7, the silicon element diffused into the MoNbTaTiW RHEA substrate at high temperatures, and the siliconized layer moved toward the substrate. The double-layer structure of the siliconized layer oxidized at 1000 °C and 1300 °C was the same as the structure of the initial siliconized layer, containing an outer high silicon content layer and an inner low silicon content layer. However, there were significant differences in the oxygen content and structure of the oxide layer after oxidation at different temperatures. The oxide layer had a single-layer structure at 1000 °C and a double-layer structure at 1300 °C. The distributions of Si, Cr, and Fe were significantly different. The outermost layer had a relatively high oxygen content at the higher oxidation temperature. It is worth noting that the contents of Cr and Fe in the siliconized layer with a low content of silicon were significantly higher than those in the siliconized layer with a high content of silicon. Some Cr and Fe elements spread to the RHEA substrate with the Si, but at a much lower diffusion rate than that of the Si element.

According to the above analysis, the initial silicide coating had a two-layer structure. After oxidation at 1000 °C, the coating had a three-layer structure of which two layers were silicide layers and one layer was an oxide layer. After oxidation at 1300 °C, the coating had four layers: two were silicide layers and two were oxide layers. Figure 8 shows a structural diagram of the coating in these three states. Based on the differences in the silicon and oxygen contents in each layer, the four layers were labeled as L1, L2, L3, and L4, where L1 represents the low silicon content siliconized layer combined with the substrate, L2 represents the high silicon content siliconized layer, L3 represents the low-oxygen-content oxide layer, and L4 represents the complete oxide layer.

As a transition layer between the substrate and the coating, the thickness of the low silicon content silicide layer (L1) seemed not to have a significant correlation with the diffusion temperature and time. The average thickness of the L1 layer in the initial coating was ~46 µm, and decreased to approximately 33 µm and 34 µm after oxidation at 1000 °C and 1300 °C, respectively. Thus, during the oxidation process, silicon continued to diffuse into the substrate, and the thickness of the siliconized layer (L2) with a high silicon content are more than doubled. The average thickness of the L2 layer of the initial coating was about 67 µm, and those of the L2 layers after oxidation at 1000 °C and 1300 °C were about 181 µm and 194 µm, respectively. According to Figure 8, it is obvious that the thickness of the oxide layer (L3 and L4) was related to the oxidation temperature, and the oxide layer became thicker at higher temperatures. After oxidation at 1000 °C, the average thickness of the oxide layer L3 was about 133 µm. After oxidation at 1300 °C, the average thickness of the oxide layer was ~169 µm, which the average thickness of the fully oxidized layer L4 was about 137 µm and that of the L3 layer was about 32 µm.

Table 2 summarizes the contents of the metal elements, silicon, and oxygen and possible phases existed in the L1, L2, L3, and L4 layers shown in Figure 8. The silicon content in L1 and L2 had specific values. In the L1 layer, the ratio of the total number of moles of metal elements to that of silicon atoms was about 5:3, while the ratio was about 1:2 in the L2 layer. The compositions of L2 in Figure 8a was disilicides of various metals, which is in consistent with the XRD pattern in Figure 2b. Therefore, it can be concluded that L2 with a high silicon content was composed of metal disilicides. Some studies have shown that the silicide of the transition layer was M_5_Si_3_ [25,26,27,28,29,30,34]. In this work, the atomic ratio of metals to silicon in the L1 layer was basically the same as that of M_5_Si_3_, indicating that the silicide in the L1 layer was M_5_Si_3_. According to the analysis by EDS (see Table 2), L3 consisted of complex metal oxides and silicon dioxide. L4 consisted of complete oxides of silicon and metal elements. In L4 layer, because silicon and all the metal elements were completely oxidized, it had the highest oxygen content.

### 3.3. Evolution of the Structure and Compositions of Silicide Coating during the Oxidation Process

Figure 9 shows a schematic diagram of the structure and composition evolution of the Si-20Cr-20Fe coating during preparation and oxidation. Based on the analysis in Section 3.1 and Section 3.2, in the initial silicide coating and the coatings after oxidation at 1000 °C and 1300 °C, L1 was M_5_Si_3_, L2 was MSi_2_, L3 was a complex oxide layer containing SiO_2_ and metal oxides, and L4 was a complete oxide layer that was composed of SiO_2_ and metal oxides.

At 1430 °C, the MoNbTaTiW RHEA substrate reacted with the molten Si-20Cr-20Fe slurry to form a double-layered silicide coating (Figure 9a,b). This silicidation reaction was quite complicated, all the metal elements including Nb, Mo, Ta, Ti, W, Cr, and Fe participated the reaction. Samosonov et al. established the relationship between the enthalpies of silicide formation and the silicon content, suggesting that for each metal atom, the enthalpy of silicide formation increased with increasing silicon content. It indicated that the silicide tended to bind more silicon and disilicide had the best stability [35]. Therefore, the surface of the initial silicide coating contained enough silicon atoms to form a disilicide layer and the reaction layer existed in the form of M_5_Si_3_ at the interface. The difference of the Gibbs free energies between MSi2 and 1/5M_5_Si_3_ (M = Ta and Ti) was very smaller than those of other elements [35]. This may lead to the appearance of Ti_5_Si_3_ and Ta_5_Si_3_ in the silicide coating. The silicide of Fe could not be detected by XRD. Nb, Cr, and Si formed a ternary silicide Cr_4_Nb_2_Si_5_ whose silicon content was between those of MSi_2_ and M_5_Si_3_. During the oxidation process of the silicide coating at high-temperature oxidation environment (Figure 9c,d), the silicon element diffused into the RHEA substrate, making the silicided layer move toward the substrate. With the increasing temperature, the degree of oxidation of metal elements was more complete and the oxide layer became thicker, and the oxide layer evaluated to a double-layer structure after oxidation at 1300 °C. There were no significant differences in the thickness, structure, and chemical compositions of the silicide layer.

## 4. Conclusions

(1) The Si-20Cr-20Fe coating was prepared on the surface of MoNbTaTiW RHEA by a fused slurry method. The initial silicide coating had a double-layer structure containing an outer layer of MSi_2_ and an inner layer of M_5_Si_3_.

(2) Under high-temperature oxidation conditions, the silicon element diffused from the silicide coating to the RHEA substrate while the oxidation of the coating occurred. After oxidation, the coating was composed of an outer oxide layer and an inner silicide layer. The silicide layer with a double-layer structure moved toward the substrate, led to the increase of its thickness.

(3) Compared with the initial silicide layer, the basic structure of silicide layer did not change significantly after oxidation. The structure and compositions of the oxide layer on the outer surface strongly depended on the oxidation temperature.

## Figures and Tables

**Figure 1 materials-13-03592-f001:**
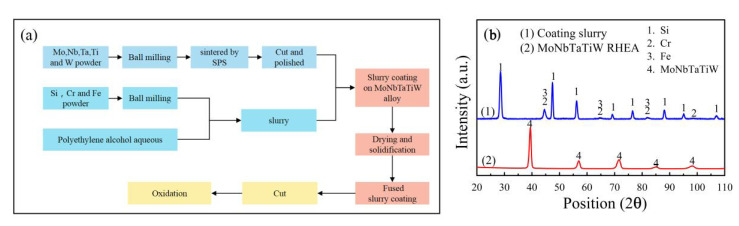
(**a**) The flowchart of the preparation process of samples and (**b**) XRD diffraction patterns of the coating slurry and the MoNbTaTiW RHEA.

**Figure 2 materials-13-03592-f002:**
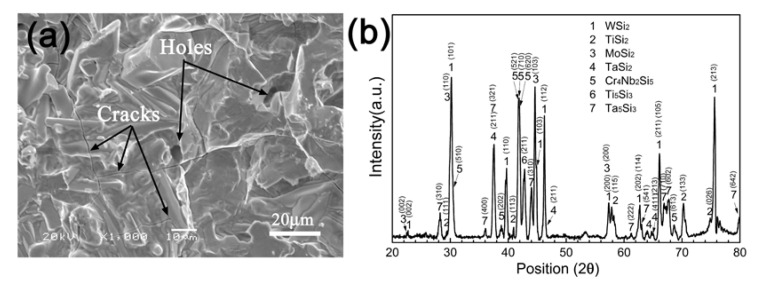
(**a**) SEM image of the surface of the silicide coating and (**b**) its XRD pattern.

**Figure 3 materials-13-03592-f003:**
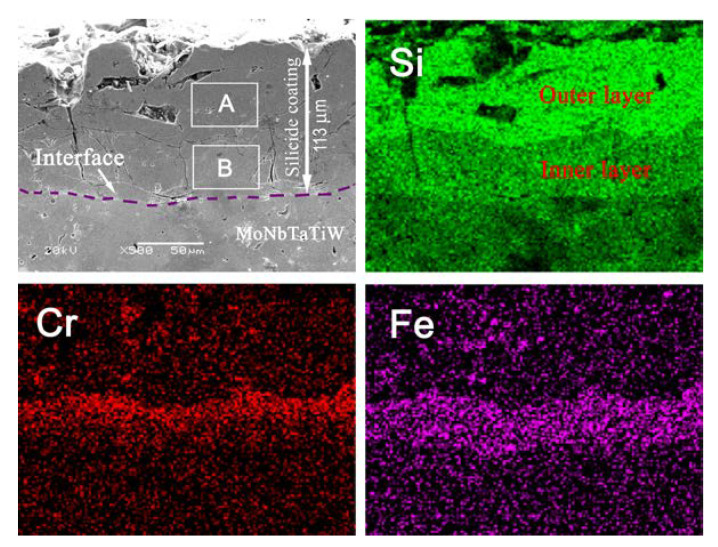
SEM image of the cross-section of the silicide coating and its energy dispersive spectroscopy (EDS) elemental mapping.

**Figure 4 materials-13-03592-f004:**
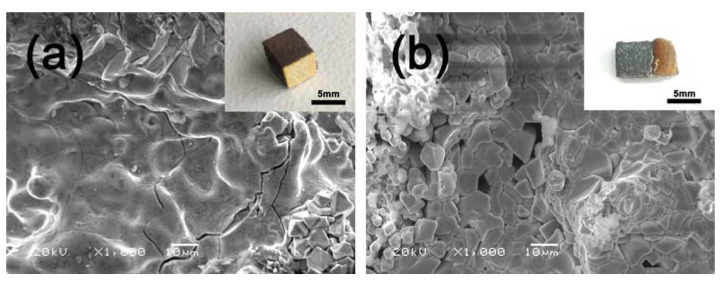
SEM images of the surfaces of the silicide coatings after oxidation at different temperatures: (**a**) 1000 °C and (**b**) 1300 °C. Inserts are the corresponding coated samples after oxidation.

**Figure 5 materials-13-03592-f005:**
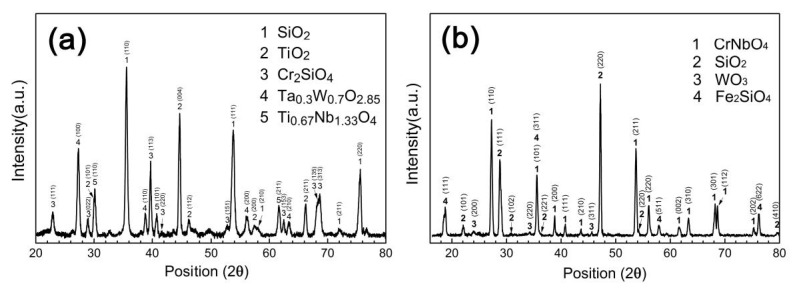
XRD patterns of the surfaces of the silicide coating after oxidation at (**a**) 1000 °C and (**b**) 1300 °C.

**Figure 6 materials-13-03592-f006:**
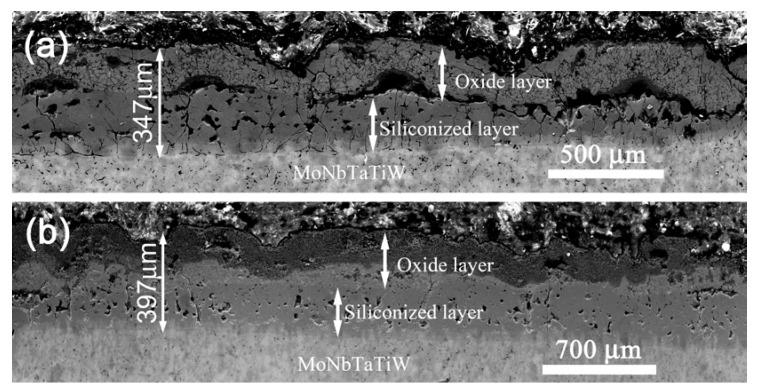
SEM images of the cross sections of the silicide coating after oxidation at (**a**) 1000 °C and (**b**) 1300 °C.

**Figure 7 materials-13-03592-f007:**
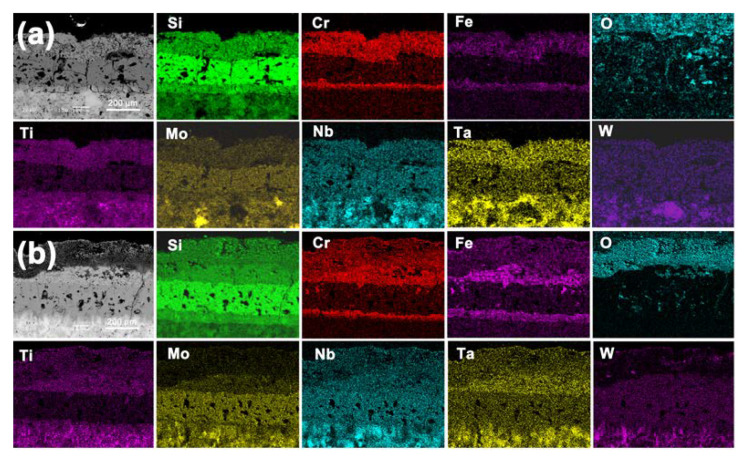
EDS mappings of the cross section of the silicide coating after oxidation at (**a**) 1000 °C and (**b**) 1300 °C.

**Figure 8 materials-13-03592-f008:**
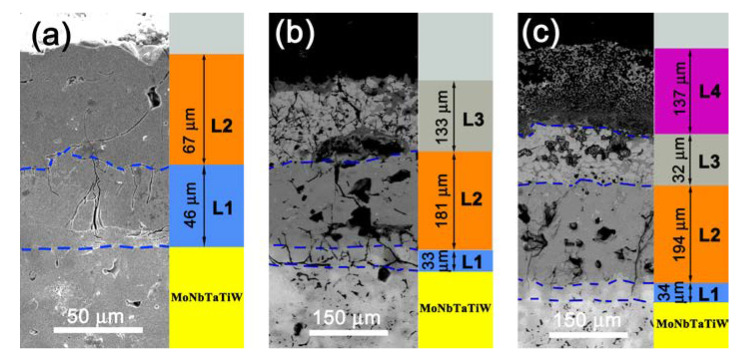
SEM images of the cross section of the coatings in different states: (**a**) initial silicide coating, (**b**) oxidized at 1000 °C, and (**c**) oxidized at 1300 °C.

**Figure 9 materials-13-03592-f009:**
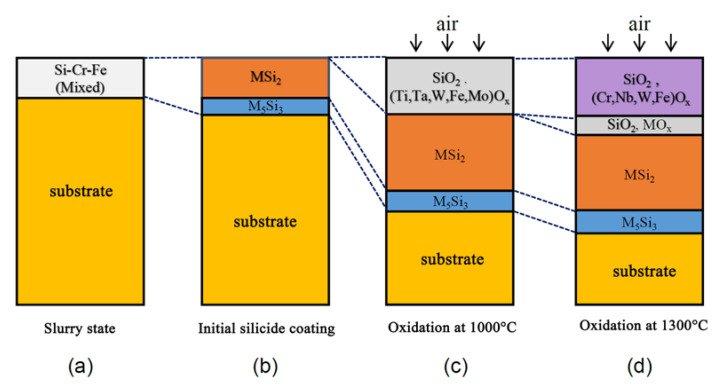
Schematic diagram of the evolution of the structure and compositions of the silicide coating during the preparation and oxidation processes: (**a**) slurry state, (**b**) initial silicide coating, (**c**) oxidized at 1000 °C, and (**d**) oxidized at 1300 °C.

**Table 1 materials-13-03592-t001:** Chemical compositions and properties of the MoNbTaTiW RHEA.

Chemical Composition, at.%					Density (g/cm^3^)	Hardness (GPa)	Yield Strength R_p0.2_ (MPa)	Peak Stress R_mc_ (MPa)	Fracture Strain ε_tc_ (%)
Mo	Nb	Ta	Ti	W					
20.0	18.5	22.3	21.1	18.1	11.6 ± 0.1	4.27 ± 0.07	1547 ± 23	1911 ± 115	11.5 ± 2.6

**Table 2 materials-13-03592-t002:** Relative contents of metal, silicon, and oxygen in the different layers measured by EDS, and possible phases for the three states: initial silicide coating, oxidized at 1000 °C, and oxidized at 1300 °C.

Layer	Initial Silicide Coating			Oxidized at 1000 °C			Oxidized at 1300 °C			Possible Phase
	M	Si	O	M	Si	O	M	Si	O	
L1	62.2	37.8	-	58.1	41.9	-	59.2	40.8	-	M_5_Si_3_
L2	33.8	66.2	-	34.2	65.8	-	34.6	65.4	-	MSi_2_
L3	-	-	-	31.3	48.1	20.6	56.9	38.8	4.3	SiO_2_, Ti_2_O, Cr_2_SiO_4_, Ta_0.3_W_0.7_O_2.85_, Ti_0.6_7Nb_1.33_O_4_
L4	-	-	-	-	-	-	14.3	18.1	67.6	SiO_2_, WO_3_, CrNbO_4_, Fe_2_SiO_4_

Note: M is the sum of the atomic ratios of all the metal elements.

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
