# Peer review of "Microstructure and Composition Evolution of a Fused Slurry Silicide Coating on MoNbTaTiW Refractory High-Entropy Alloy in High-Temperature Oxidation Environment"

_materials, 2020, doi:10.3390/ma13163592_

Round 1

Reviewer 1 Report

In their manuscript the authors investigate improvements of the high-temperature oxidation stability of RHEAs of MoNbTaTiW. Prior to publication some points should be addressed, that can enhance the readability of the paper.

1.) Line 1.64: Why does the preferred coating change for Nb, W or Ta containing AlCrMoTi-X alloys from a materials science point of view?

2.) What is the role of initial cracks and holes in the silicide coating of the RHEAs in the oxidation process? Does the oxide penetration depths change close to the cracks?

3.) Line 3.58: Why are Cr and Fe enriched in the inner layer – which elements diffuse faster, Cr/Fe or RHEA elements?

4.) Can the authors speculate about the role of the plane-parallel cracks between oxide and silicide layers formed at 1000°C for longer oxidation times? The cracks act as diffusion barriers – how would the oxide structure change after longer time?

5.) Fig. 7: The same colours should be used for the same chemical elements.

6.) Table 2: According to the text, M in L1 and M in L2 does not subsume the same chemical elements, but in L1 M is only Ta and Ti. This should be noted in the table. Why do W and Mo degrade in layer L1?

Reviewer 2 Report

The text is worth publishing. However, its layout should be improved and minor remarks by the evaluator should be taken into account.
1. We do not include general information in the summary. The summary is intended to summarize the value of the research and results. Please edit them.
2. Keywords should reflect the substantive sense of the publication. We put them in order from most important to less important.
3. The results of the works cited should be described in more detail in the literature review. Each of them concerns a different aspect of the undertaken research topic.
4. Text should be checked in editor, commas, spaces, etc.
5. The composition of the MoNbTaTiW coating should not be emphasized.
6. Table 1 contains incorrect markings and the number format. In addition, the parameter designations should be included.
7. In the Materials and methods chapter - put a flowchart of the sample process.
8. In many places there are erroneous measurements of parameter values.
9. Please explain in the text why ingredient powders of the given sizes were used. Why was such a composition and procedure for the formation of the coating used.
10. The scale in image no. 2a is unreadable.
11. In Figure 3, the components Si, Cr, Fe should be better described.
12. Please do not include references to a), b) etc. on the drawings - they are illegible.
13. Please explain the causes of micro-cracks in the coating in more detail.
14. In Figure 6 phases of the layers are not visible - if possible, please improve the sharpness of the image. The scale is unreadable.
15. In Figure 7, please improve the readability of the component density distribution.
16. Please describe the shell XRD analysis process in more detail.
17. Please describe the layers L1, L2, L3 etc. in more detail.
18. Figure 9 shows the differences in oxidized at 1000 oC, and oxidized at 1300 oC in more detail.
19. The conclusions are too general. It should be based on the results of quantitative and qualitative analysis.

Reviewer 3 Report

The authors have used a fused slurry method to coat Si20Cr-20Fe alloy over the MoNbTaTiW refractory high entropy alloy in a view to improving the oxidation resistance at high temperature. This work can be interesting for the powder metallurgy community for improving the surface properties of the RHEAs. However, some issues need to be addressed.

(1) The preparation of substrate RHEA is not given in detail. Please add.

(2) Line 98-99 mentions that HEAs are prepared by spark plasma sintering. Further line 104- mentions vacuum hot pressing? Please check if these two processes are different.

(3) Line 144-147, I see some strange underlined text in blue. What is this?
Similarly, for lines 150-170.

(4) Figure 4: It would be better if the authors compare uncoated RHEAs with silicide coated RHEAs. Also, please make a comparison of substrate RHEA with silicide coated RHEA and comment on the surface protection efficiencies, for example.
